# Fatigue Properties of AZ31B Magnesium Alloy Processed by Equal-Channel Angular Pressing

**Ryuichi Yamada [1,*], Shoichiro Yoshihara [2] and Yasumi Ito [1]**

[1] Graduate Faculty of Interdisciplinary Research Faculty of Engineering, Mechanical Engineering (Mechanical Engineering), University of Yamanashi, 4-3-11 Takeda, Kofu-shi 400-8511, Japan; yasumii@yamanashi.ac.jp

[2] Department of Engineering and Design, Shibaura Institute of Technology, 3-9-14 Minato-ku, Tokyo 108-8548, Japan; yoshi@shibaura-it.ac.jp

[*] Correspondence: ryamada@yamanashi.ac.jp; Tel.: +81-55-220-8091

**Abstract:** A stent is employed to expand a narrowed tubular organ, such as a blood vessel. However, the persistent presence of a stainless steel stent yields several problems of late thrombosis, restenosis and chronic inflammation reactions. Biodegradable magnesium stents have been introduced to solve these problems. However, magnesium-based alloys suffer from poor ductility and lower than desired fatigue performance. There is still a huge demand for further research on new alloys and stent designs. Then, as fundamental research for this, AZ31 B magnesium alloy has been investigated for the effect of equal-channel angular pressing on the fatigue properties. ECAP was conducted for one pass and eight passes at 300 °C using a die with a channel angle of 90°. An annealed sample and ECAP sample of AZ31 B magnesium alloy were subjected to tensile and fatigue tests. As a result of the tensile test, strength in the ECAP (one pass) sample was higher than in the annealed sample. As a result of the fatigue test, at stress amplitude $\sigma_a$ = 100 MPa, the number of cycles to failure was largest in the annealed sample, medium in the ECAP (one pass) sample and lowest in the ECAP (eight passes) sample. It was suggested that the small low cycle fatigue life of the ECAP (eight passes) sample is attributable to severe plastic deformation.

**Keywords:** magnesium alloy; fatigue; equal-channel angular pressing; grain refinement; S–N curve; stent

## 1. Introduction

Ischaemic heart disease is the world's biggest killer, accounting for a combined 9.4 million deaths in 2016 [1]. This disease has remained the leading cause of death glob-ally in the last 15 years [1]. A stent, as a countermeasure against ischaemic heart disease, is employed to expand a narrowed tubular organ, such as a blood vessel. A stent is a tiny tube that a doctor places in an artery or duct to help keep it open and restore the flow of bodily fluids in the area. Stents help relieve blockages and treat narrow or weakened arteries. Despite this, the persistent presence of a metallic stent yields several problems of late thrombosis, restenosis and chronic inflammation reactions. Conventionally, an austenite-based stainless steel that is a general stainless steel, such as SUS316, SUS316 L, etc., is used for the material of a stent for vasodilatation [2]. Once the stainless steel stent is implanted, it will remain in an artery permanently because of its excellent corrosion resistance. In some cases, restenosis may occur [3]. Restenosis is when too much tissue grows around the stent. This could narrow or block the artery again. In general, a stent should be removed within 6 to 9 months. Stent removal surgery is expensive and physically demanding for the patient. Biodegradable magnesium stents have been introduced to solve these problems. There is still a huge demand for further research on new alloys and stent design [4,5]. Ideally, implanted stents can maintain their mechanical integrity during the healing of the vessel wall and then dissolve after healing. The mechanical strength and properties of magnesium are suitable for biodegradable implants, especially for stent

application. Magnesium is bi-ocompatible because it is essential for several biological reactions and as a co-factor for enzymes. However, magnesium degradation is accelerated in chloride-abundant en-vironments, such as human body fluids. Therefore, magnesium must be modified to improve its corrosion resistance. In addition, magnesium-based alloys suffer from poor ductility [6] and lower than desired fatigue performance. Further, the great success of stents in treating cardiovascular disease is undermined by their long-term fatigue failure. Pulsatile pressure and repetitive mechanical forces within the coronary artery may result in fatigue fracture after stent implantation, particularly in patients with complex coronary disease [7]. It has also been reported that stents implanted near the heart have a higher probability of fatigue failure. In this research group, it has been found that the corrosion rate of magnesium alloy becomes larger under pulsating flow [8]. If the fatigue properties can be improved, there is a possibility of reducing the corrosion rate. These aspects make fatigue properties an important attribute of cardi-ovascular stents. In this study, we focused on the possibility of improving the strength and fatigue properties simultaneously by grain refinement. Then, as fundamental re-search for this, AZ31 B magnesium alloy has been investigated for the effect of equal-channel angular pressing on tensile and fatigue properties.

## 2. Materials and Methods

### 2.1. Specimens

The chemical composition (in mass %) of the AZ31 B magnesium alloy made by MACRW Co., Ltd. (Shizuoka, Japan) used in this study is listed in Table 1. The AZ31 B alloy was hot extruded to a rod with a diameter of 6 mm and then cut into pieces with a length of 60 mm. ECAP was conducted on the as-extruded material through a die with an internal angle $\varphi$ of 90° between the vertical and horizontal channels and a curvature angle $\psi$ of 90° (Figure 1). As a lubricant, we used molybdenum disulphide ($MoS_2$) manufactured by Sumico Lubricant Co., Ltd. (Tokyo, Japan). Conditions of ECAP are presented in Table 2. ECAP was conducted for 1 pass and 8 passes, which creates an equivalent strain of 0.91 during one passage through the die as shown in Equation (1) [9].

$$\varepsilon_N = \left( \frac{N}{\sqrt{3}} \right) [2cot\left\{ \left( \frac{\varphi}{2} \right) + \left( \frac{\psi}{2} \right) \right\} + \psi cosec\left\{ \left( \frac{\varphi}{2} \right) + \left( \frac{\psi}{2} \right) \right\}] \qquad (1)$$

$\varepsilon_N$ is the strain of the material after ECAP for $N$ passes and $N$ is extrusion pass number. For comparison, the annealed sample for AZ31 B consisted of hold for 1 h at 450 °C, furnace cooling. ECAP-8 was used because the maximum number of passes that would not cause cracking under the conditions of this experiment was 8 passes. Furthermore, ECAP-8 p samples were found to have inferior fatigue properties compared to annealed samples [10]. ECAP-1 p was also added to investigate the effect of reducing the number of passes for ECAP. Three distinct kinds of test pieces, i.e., ECAP-1 p, ECAP-8 p and annealed, had commonly equiaxed microstructures (Figure 2). The average grain sizes of annealed and ECAP-8 p samples were confirmed by optical microscope observation to be 40 μm and 6–7 μm, respectively. ECAP-1 p had a bimodal structure, and it was determined that accurate grain size could not be calculated from the average grain size. It was the optical microscope structure in which coarse grains of 30–40 μm and fine grains of less than 10 μm were mixed.

**Table 1.** Composition of the specimens in mass %. Bai. in Mg is an abbreviation for balance.

|        | Al   | Zn   | Mn   | Si   | Fe    | Mg   |
|--------|------|------|------|------|-------|------|
| AZ31 B | 3.14 | 1.11 | 0.33 | 0.02 | 0.001 | Bal. |

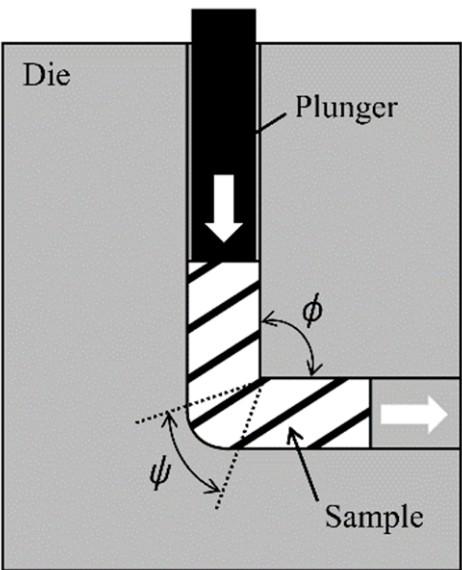

**Figure 1.** Schematic diagram of ECAP processing.

**Table 2.** Composition of the specimens in mass%.

| Processing Temperature | 300 °C |
|---|---|
| Processing rate | 3 mm/min |
| Hole diameter of die | 6 mm |
| Channel angle $\varphi$, $\psi$ | 90°, 90° |
| Number of passes (Rout Bc) | 1 pass 8 passes |

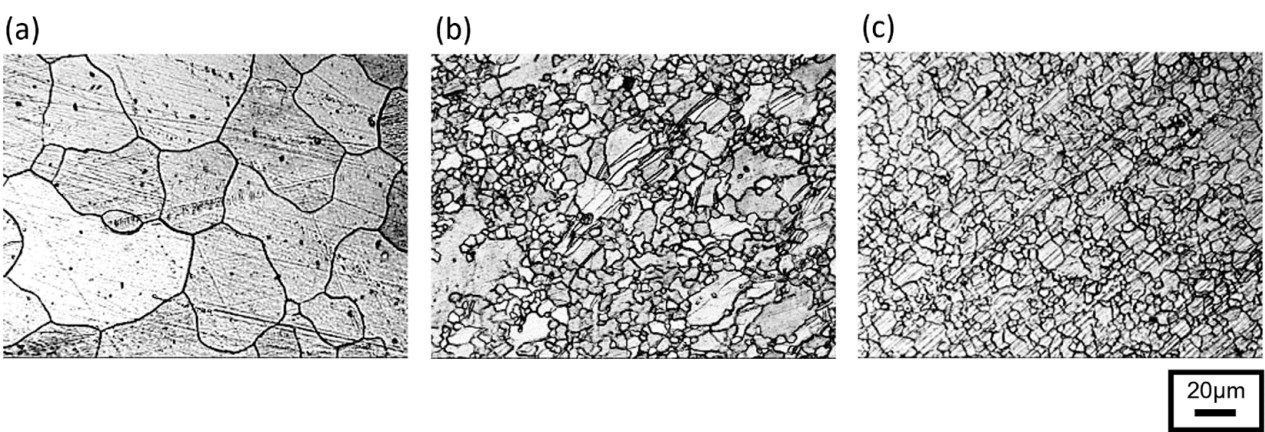

**Figure 2.** Optical micrographs of AZ31 B alloy. (**a**) Annealed, (**b**) ECAP-1 p and (**c**) ECAP-8 p.

### 2.2. Experimental Procedure

From the hot-extruded rod or ECAPed sample, tensile and fatigue test pieces were produced by machining. The gauge length and diameter of tensile and fatigue test pieces were 15 mm and 3 mm, respectively. The round-rod test pieces were turned on a lathe and manually mechanically polished and buffed to a mirror finish. The tensile test was carried out on one specimen of each type at an initial strain rate of $1.5 \times 10^{-3}$ s$^{-1}$ at room temperature. ECAP and tensile tests were performed by an Autograph AG-250 kNG, a precision universal tensile tester manufactured by SHIMADZU CORPORATION (Kyoto,

Japan). Fatigue test was carried out on one specimen of each type in a sinusoidal stress wave of a frequency of 30 Hz with a stress ratio of R (= $K_{min}/K_{max}$) = 0.1 at room temperature, followed by fractographic observation by a scanning electron microscope (SEM, JSM-7100 F by JEOL Ltd., Tokyo, Japan). The fatigue testing machine was an ElectroPuls E10000 Linear-Torsion all-electric dynamic test instrument manufactured by INSTRON Japan (Kanagawa, Japan). Residual stress measurement was carried out by X-ray diffraction (XRD, SmartLab by Rigaku Corporation, Tokyo, Japan) on the gripping surface of the test piece after the fatigue test. XRD was employed to measure the residual stress of the rods, in which Cr Kα radiation was used as the X-ray source. X-ray generator was attached by a collimator, which could carry out the incident beam of 1 mm spot sizes.

## 3. Experimental Results

### 3.1. Tensile Test

Figure 3 shows nominal stress vs. strain curves for the three specimens. Strength in the ECAP-1 p sample was higher than in the annealed sample. The tensile strength of ECAP-1 p was 321 MPa, which was slightly improved by about 1.2 times compared to the annealed sample. In contrast, elongation to failure was larger in the ECAP-8 p sample than in the annealed sample. In spite of processing more, ECAP-8 p had lower tensile strength and lower yield strength than ECAP-1 p. The elongation to failure of ECAP-8 p was 23%, which was improved by about 1.4 times compared to the annealed sample. Tensile properties obtained from tensile tests for the three specimens are summarized in Table 3.

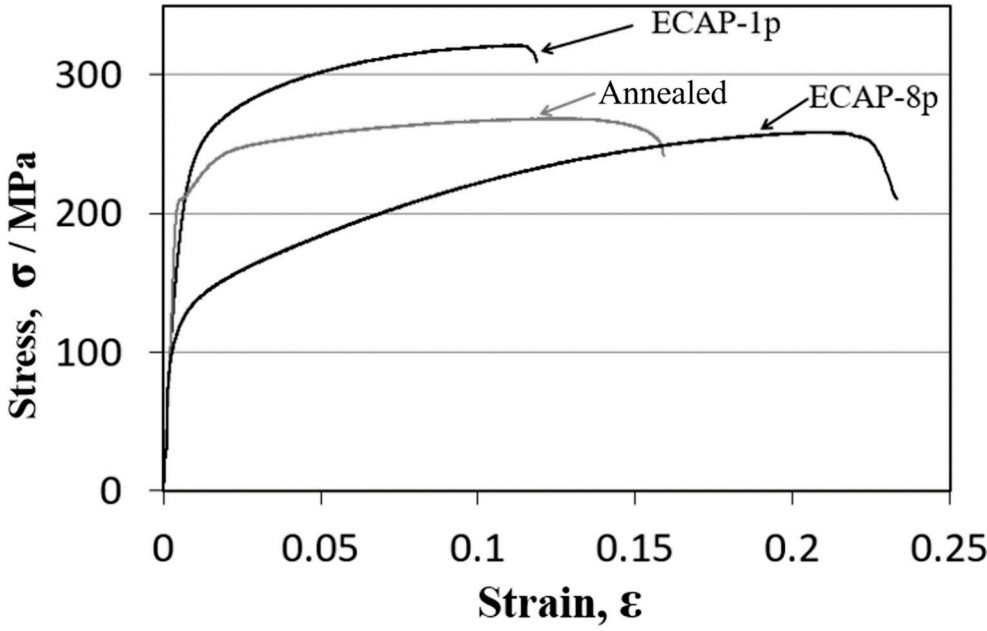

**Figure 3.** Nominal stress vs. strain curves for annealed, ECAP-1 p and ECAP-8 p samples of AZ31 B alloy.

**Table 3.** Tensile properties (ultimate tensile strength $\sigma_u$, yield strength $\sigma_y$, elongation to failure $\delta$) of AZ31B alloy.

| | Tensile Strength $\sigma_u$ MPa | Yield Strength $\sigma_y$ MPa | Elongation $\delta$ % |
|---|---|---|---|
| AZ31B Annealed | 269 | 217 | 16 |
| AZ31B ECAP-1p | 321 | 200 | 12 |
| AZ31B ECAP-8p | 259 | 126 | 23 |

*3.2. Fatigue Test*

As a result of the fatigue test in Figure 4, at stress amplitude $\sigma_a$ = 100 MPa number of cycles to failure was largest in the annealed sample, medium in the ECAP-1 p sample and lowest in the ECAP-8 p sample. At stress amplitude $\sigma_a$ = 80 MPa, the number of cycles to failure was largest in the ECAP-1 p sample, medium in the annealed sample and lowest in the ECAP-8 p sample. It was confirmed that grain refinement improved fatigue life at low stress amplitude. When the stress amplitude $\sigma_a$ exceeds 80 MPa, the annealed sample has the best fatigue properties and the ECAP-8 p sample has the worst fatigue properties, suggesting that among tensile properties, strength is more effective in improving fatigue properties than ductility. And since the fatigue properties of the annealed sample are superior to those of the ECAP-1 p sample, it is suggested that increasing the yield strength rather than the tensile strength is effective in improving the fatigue properties.

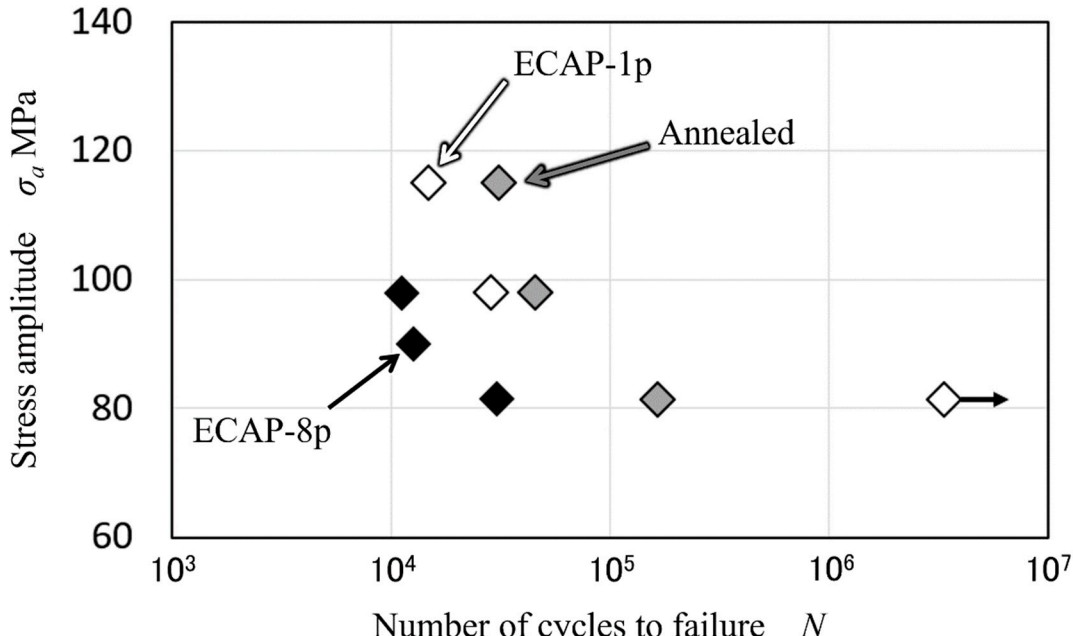

**Figure 4.** S–N curves for annealed, ECAP-1 p and ECAP-8 p samples of AZ31 B alloy.

Figure 5 shows the SEM images of fracture surface at stress amplitude $\sigma_a$ = 100 MPa. From this, many fatigue crack initiation sites were located at each fracture surface. In all fracture surfaces, the crack growth area extends radially from the starting point of the fracture indicated by the arrow to the area delineated by the dashed line. In ECAP-1 p and ECAP-8 p, which are shown in Figure 5b,c, the crack growth areas from different fracture initiation points overlap each other. In ECAP-8 p, which had a large number of passes, the number of starting points for fracture was four, which was larger than the two in ECAP-1 p, suggesting early fatigue fracture. In the annealed sample, the starting points of the destruction overlap, however they are not completely connected. Therefore, it is inferred that the fatigue life of ECAP-1 p and ECAP-8 p was reduced by the sudden decrease in the cross section of the parallel section of the fatigue specimen when the crack growth areas overlapped each other. Although it is qualitative, the crack growth area ratio is larger in the annealed and ECAP-1 p sample than in ECAP-8 p. Therefore, it is possible that the crack growth area ratio of the fracture surface increases with the superior fatigue property.

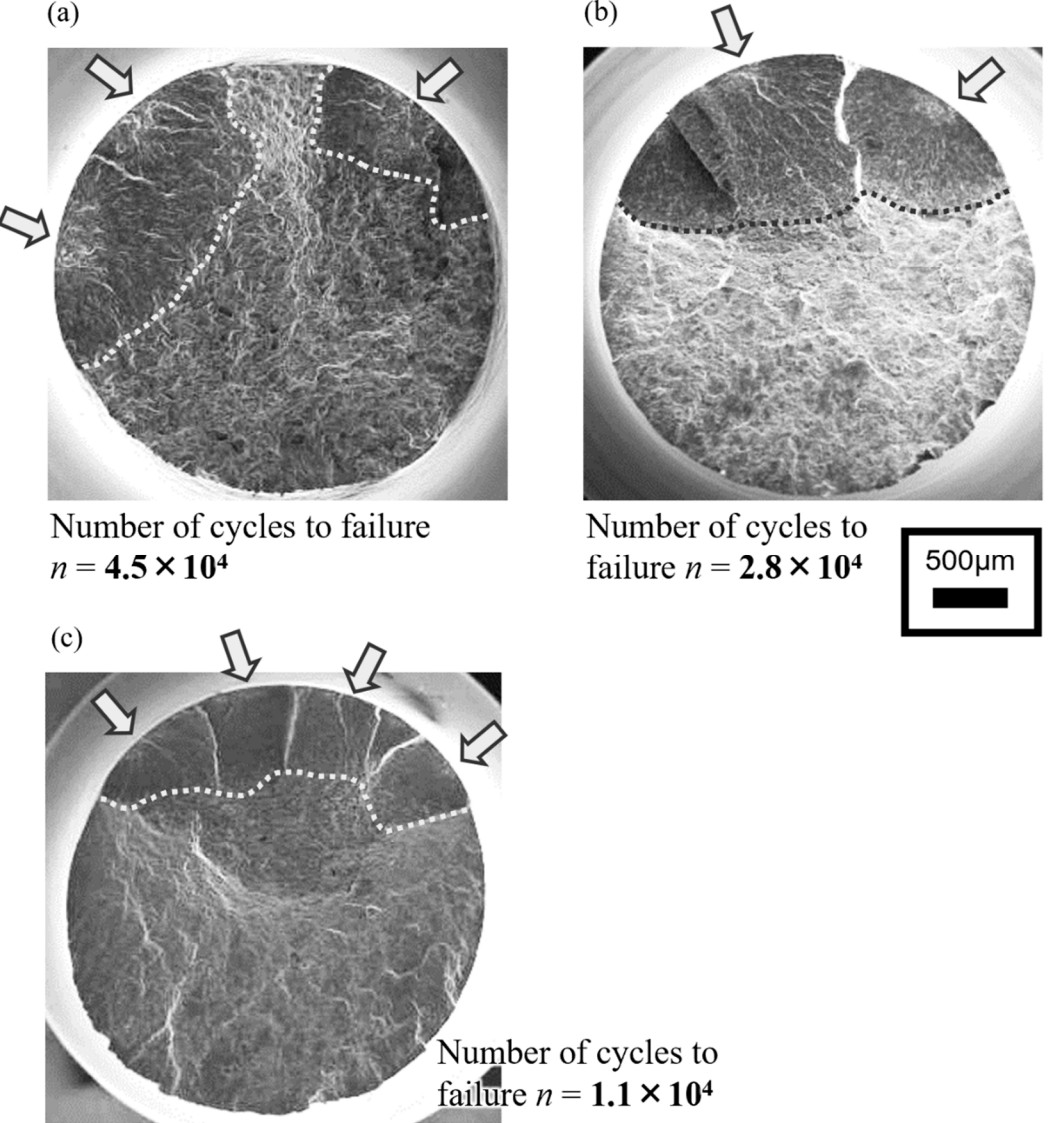

**Figure 5.** SEM images of fracture surfaces of annealed, ECAP-1 p and ECAP-8 p samples of AZ31 B alloy at stress amplitude $\sigma_a$ = 100 MPa. (**a**) Anneal, (**b**) ECAP-1 p, (**c**) ECAP-8 p.

Figure 6 is a SEM image which enlarged fatigue crack initiation sites; fracture surfaces are found mainly to consist of transgranular quasi-cleavage. Annealed and ECAP-1 p samples showed a fracture surface with severe surface undulations as if plastic deformation had occurred to a large extent, which could be seen as twinning. No twinning-like structures were observed in ECAP-8 p. It is inferred that the twinning deformation at the starting point of fracture in the annealed and ECAP-1 p samples resist fatigue fracture and the fatigue life is longer than that of ECAP-8 p.

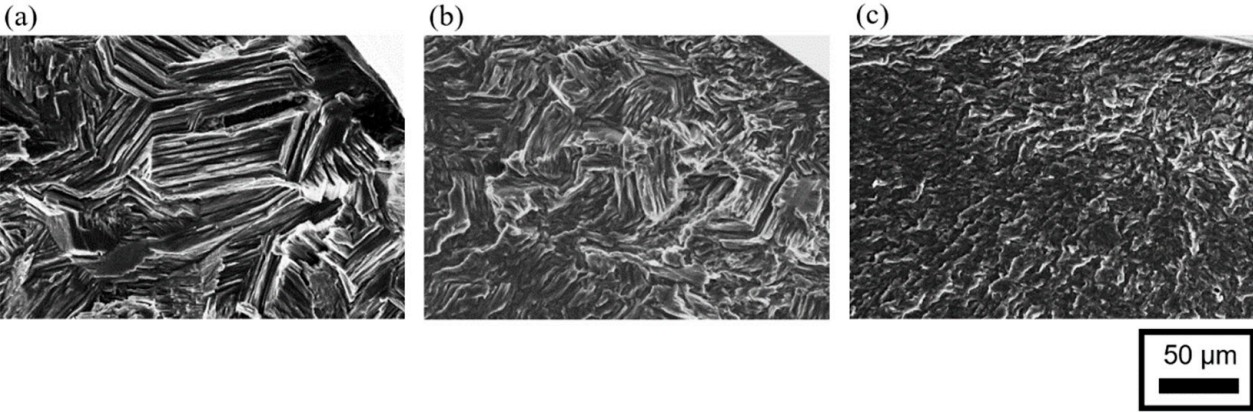

**Figure 6.** High-magnification SEM images of fracture surfaces of annealed, ECAP-1 p and ECAP-8 p samples of AZ31 B alloy at stress amplitude $\sigma_a$ = 100 MPa. (**a**) Anneal, (**b**) ECAP-1 p, (**c**) ECAP-8 p.

## 4. Discussion

In a related study, it was found that the fatigue properties of Mg–Al alloys decreased as the number of passes in ECAP increased [11]. Although it is AZ80 magnesium alloy, the fatigue properties of the six passes material decreased, and the two passes material was superior in fatigue properties [11]. It has been reported that the fatigue properties of AZ31 magnesium alloy, which is close in composition, were improved by ECAP processing, but the number of passes was small, four passes [12]. Therefore, the decrease in the fatigue properties of ECAP-8 p in the present experimental results may be due to the increase in the number of passes. Residual stress measurement was performed to consider the cause of the deterioration of fatigue properties due to ECAP processing at high stress amplitude (Figure 7). For all three specimens, the residual compressive residual stress after the fatigue test was measured. The annealed sample seems to have had little residual stress before the fatigue test. Since the compressive stress value is lower in the annealed sample, it can be inferred that ECAP-1 p and ECAP-8 p had tensile stress generated by ECAP processing. ECAP-1 p and ECAP-8 p showed tensile residual stress, which increased with the number of passes, and ECAP-8 p was considered to have the earliest fatigue failure. It has also been reported that both strength and elongation can be increased by artificial aging heat treatment after ECAP processing in AZ91 magnesium alloy [13]. As future work, it is necessary to verify whether artificial aging of ECAP-1 p can exceed the fatigue properties of annealed samples. The mechanical properties of ECAP samples with varying number of passes between two and seven should also be continued to be investigated.

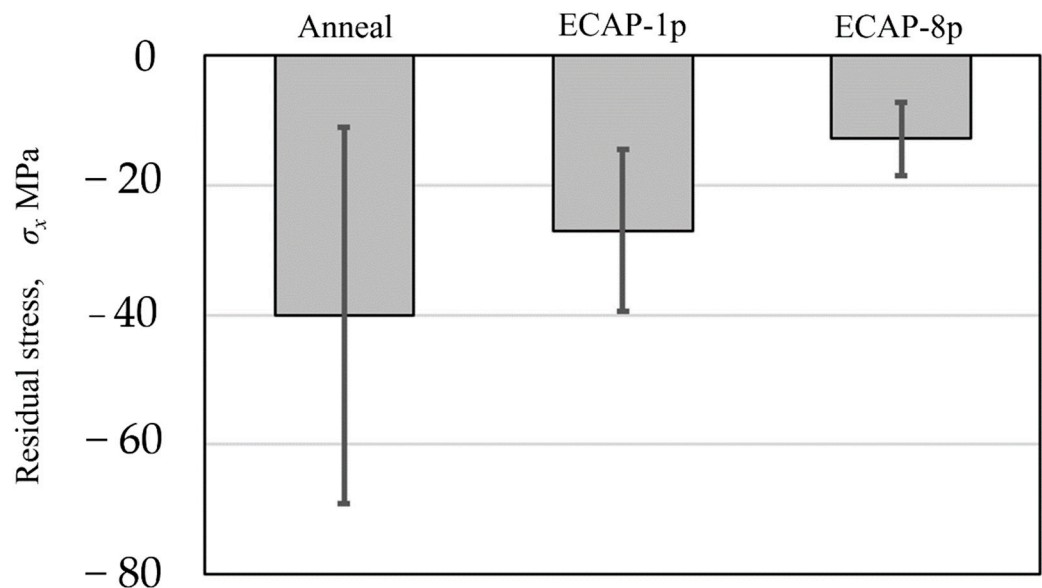

**Figure 7.** Residual stress of annealed, ECAP-1 p and ECAP-8 p samples of AZ31 B alloy measured by XRD after fatigue test at stress amplitude $\sigma_a$ = 100 MPa.

## 5. Conclusions

The effects of varying the number of passes on tensile and fatigue properties of AZ31 B magnesium alloy under the present experimental conditions were investigated when equal-channel angular pressing was performed. Tensile strength in ECAP-1 p was higher than in the annealed sample. In contrast, elongation to failure was larger in the ECAP-8 p sample than in the annealed sample. From the fatigue test, at stress amplitude $\sigma_a$ = 100 MPa, number of cycles to failure was largest in the annealed sample, medium in the ECAP-1 p sample and lowest in the ECAP-8 p sample. At stress amplitude $\sigma_a$ = 80 MPa, number of cycles to failure was largest in the ECAP-1 p sample and medium in the annealed sample. It was found that grain refinement by ECAP was effective in improving fatigue life depending on the value of stress amplitude. Strength is more effective than ductility in improving fatigue properties, and it is considered important to increase the yield strength. It was suggested that the low fatigue life of the ECAP-8 p sample is attributable to tensile residual stress in severe plastic deformation.

**Author Contributions:** Conceptualization, R.Y. and S.Y. and Y.I.; methodology, R.Y.; writing—original draft preparation, R.Y.; supervision, R.Y. and S.Y. and Y.I.; project administration, R.Y.; funding acquisition, S.Y. All authors have read and agreed to the published version of the manuscript.

**Funding:** This research was funded by JSPS KAKENHI, grant number JP19K04095.

**Institutional Review Board Statement:** Not applicable.

**Informed Consent Statement:** Not applicable.

**Conflicts of Interest:** The authors declare no conflict of interest.

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
