# Peer review of "Fatigue Properties of AZ31B Magnesium Alloy Processed by Equal-Channel Angular Pressing"

_metals, doi:10.3390/met11081191_

Round 1

Reviewer 1 Report

The references should eb adjusted as per mdpi requirements

“In order to elucidate the mechanism of fatigue in the AZ31B magnesium alloy” this apply for this specific manufacturing process and not for all processing condition. So, I advice to re write this statement

Very interesting but a ref is required for “combined 9.4 million deaths in 2016”

A ref is required “In some cases, restenosis may occur.”

A ref is required “magnesium-based alloys suffer from poor ductility”

Can you provide the supplier for “Molybdenum disulphide (MoS2)”

Why “ECAP-8p” and not more or less?

“The round rod test pieces were mirror-finished” manually or automatically ?

A ref is required “caused by recrystallization because the processing temperature” otherwise is just a  speculation

Figure 5 and 6 were presented in but not discussed or interpreted in text

There is need a section of discussion to interpret these results against literature data

Overall more details will make this work more attractive

Author Response

Thank you very much for providing important comments.  We are thankful for the time and energy you expended.  Please see the attachment.

Reviewer 2 Report

The introduction provides a good, generalized background of the topic (summarizes recent research related to the topic, and gives a clear idea of the target readership, why the research was carried out and the novelty and topicality of the manuscript). The introduction is short but concise.

The objective is clearly defined and highlights gaps in current understanding in current knowledge are presented. The results are presented succinctly.

The literature cited is relevant to the study.

Overall, the authors makes a good contribution to the field, the correct statistics were used and the interpretation of the data makes sense. The results are interpreted appropriately. The paper is clear and concise. In my humble opinion, the conclusions could be broader, richer in general analysis and specific.

Author Response

(The authors gave the same response as above.)

Reviewer 3 Report

General comments

The article examines the effect of Equal-Channel Angular Pressing on the fatigue properties of AZ31B magnesium alloy. 2 types of ECAP (1, 8 passes) were performed and compared with the alloy after the annealing process. The tensile test results showed that the sample after one pass of ECAP had the best tensile properties, while the samples after eight passes had better plastic properties. However, the results of the fatigue tests showed that the number of cycles to failure was largest in the annealed sample, medium in the ECAP (1 pass) sample, and lowest in the ECAP (8 passes) sample. It was confirmed that grain refinement improved fatigue life at low-stress amplitude. Moreover, the authors performed the measurements of the residual stresses using the XRD method, which shows that the alloy after annealing was characterized by the lowest values ​​of the residual stresses. The ECAP method generates additional stresses that affect the fatigue strength of the alloy.

The paper is obviously of interest to researchers working in this field. However, it requires minor revisions:

  1. The introduction is based on only seven articles. In my opinion, it requires a significant extension and analysis of the latest literature.
  2. In the research method, the authors describe very briefly all the tests performed. They do not specify the number of repetitions. In addition, the XRD research methodology is included in the results chapter, and please move it to the methodology.
  3. How was the chemical composition (Table 1) of the alloy tested?
  4. Figure 3 shows the tension curves. Are they sample tests or their average value? How many tests were performed?
  5. In Figure 4, there is no marked standard deviation for each of the samples. The question is the same as in the previous point. How many tests were performed for each case?

After expanding the literature introduction and introducing changes in the methodology and research results, I believe that the article will be acceptable for publication.

Author Response

(The authors gave the same response as above.)

Round 2

Reviewer 1 Report

-

Author Response

T